



# Investigating cold based summit glaciers through direct access to basal ice: A case study constraining the maximum age of Chli Titlis glacier, Switzerland

Pascal Bohleber[1,2], Helene Hoffmann[2], Johanna Kerch[2,3], Leo Sold[4], and Andrea Fischer[1]

[1]Institute for Interdisciplinary Mountain Research, Austrian Academy of Sciences, Innsbruck, Austria
[2]Institute of Environmental Physics, Heidelberg University, Heidelberg, Germany
[3]Alfred Wegener Institute Helmholtz Center for Polar and Marine Research, Bremerhaven, Germany
[4]formerly at: Department of Geosciences, University of Fribourg, Fribourg, Switzerland

*Correspondence to:* Pascal Bohleber (Pascal.Bohleber@iup.uni-heidelberg.de)

**Abstract.** Cold glaciers at the highest locations of the European Alps have been investigated with great success by drilling ice cores to retrieve their stratigraphic climate records. Findings like the Oetztal ice man have demonstrated that small ice bodies at summit locations of comparatively lower altitudes may also contain old ice if comprising ice frozen to the underlying bedrock. In this case, constraining the maximum age of their basal ice part may help to identify past periods with minimum ice extent in the Alps. Facing ongoing warming and recent years with extremely negative glacier mass balance, these sites may not preserve their unique climate information for much longer, however. Since sampling and dating the lowermost ice is essential, and usually requires substantial logistical (drilling) effort, we utilize here the direct access to basal ice offered by an existing ice cave at Chli Titlis (3030 m), Central Switzerland. Our dedicated approach comprises a combination of standard glaciological tools with the analysis of the isotopic and physical properties and sophisticated radiocarbon dating techniques. By this means we demonstrate that, in comparison to an earlier study at Chli Titlis, stagnant cold basal ice conditions still exist fairly unchanged more than 25 years after the pioneering exploration. Our radiocarbon analysis constrains the maximum age of the ice at Chli Titlis to about 5000 years before present. By this means, the approach presented here will contribute to a future systematic investigation of cold-based summit glaciers also targeting the Eastern Alps.

## 1 Introduction

Glaciers in high mountain environments are able to archive climate signals in regions and altitudes where other proxy records are scarce. Non-temperate sedimentary glaciers (i.e. archiving snow on a quasi-continuous basis) can hold past climate and environmental signals that can be retrieved by drilling ice cores. In a number of pioneering studies it was already shown that, given the climate conditions and altitude range of the European Alps, cold firn and ice areas suitable for ice core studies are located in the uppermost summit ranges, mostly in the Western Alps (e.g. Haeberli, 1976; Oeschger et al., 1977; Haeberli and Alean, 1985). In comparison, summit glaciers of lower altitudes received less attention regarding their role as climate archives, until Haeberli et al. (2004) proposed and performed initial investigations of cold ice in detail for the European Alps. Among other findings, the discovery of the Oetztal ice man at Tisenjoch (3210 masl) and the subsequent dating to more





than 5000 years before present (Baroni and Orombelli, 1996; Kutschera and Müller, 2003) demonstrated that old ice can be preserved at comparatively lower altitudes under certain conditions. First and foremost these conditions require little or no ice flow, as favored through locations near ice divides, certain bedrock geometries (e.g. depressions) and, most importantly, persistent sub-zero englacial temperatures ensuring that the ice is frozen to the underlying permafrost bedrock.

While techniques such as the dendro-chronological analyses of formerly glacier-buried tree parts usually provide evidence of glacier fluctuations at lower elevations (i.e. in the vicinity of the former glacier tongue) (Ivy-Ochs et al., 2009; Joerin et al., 2008; Nicolussi and Patzelt, 2000), the investigation of summit glaciers promises important complementary palaeo-climatic information on warm periods involving minimum ice extents in the Alps. This would also provide additional constraint to the question if today's glacier covered highest elevation have been ice free during the Holocene. At the same time, the current

warming conditions pose an immediate risk of losing this archive (c.f. the discussion of Haeberli et al. (2004) regarding the strong impact of the 2003 summer on Piz Murtel). Promising candidates for holding cold basal ice can be identified based on glacio-meterological parameters including mean annual air temperature, aspect and snow accumulation. Evidently, obtaining access to the basal ice parts is essential in this context, i.e. for direct measurements of englacial temperature, basal ice sampling and subsequent dating. A key concept here is that constraining the age of the (stagnant) basal layer at the summit

may indicate the maximum age of these ice bodies. Since the stratigraphy of the expected glacier types (usually mostly made from congelation ice) cannot be expected to include layers of every single year, conventional dating methods like annual layer counting are severely hampered and, as a result, age constraints must be obtained mostly from radiometric methods (May, 2009). Novel developments in adapting and refining radiocarbon techniques for microscopic organic material from glacier ice (Uglietti et al., 2016; Hoffmann, 2016) arrive just-in-time to offer an indispensable dating tool in this context. This is

especially the case since the expected glacier age falls within the age range for the application of the radiocarbon technique, e.g. as indicated by the dating of the Oetztal ice man.

Here we report on an investigation designed as a pilot study to a systematic investigation of cold-based summit glaciers in the Eastern Alps. Combining glaciological surveying with radiocarbon dating of ice samples offers constraining the glacier age, especially if direct access to the lowermost ice section can be obtained for sampling at large volume, e.g. through an ice cave

(Ødegård et al., 2017). This is especially important for application of the $^{14}$C technique, which is in ice core science often hampered by limited sample sizes and low organic carbon concentrations. We additionally integrated the analysis of stable water isotopes and physical ice properties to detect stagnant basal ice conditions, and selected Chli Titlis glacier, located at 3030 m asl in the central Swiss Alps as the target for this investigation. This choice was motivated by considering that at Chli Titlis i) direct access to basal ice can be obtained at low logistical cost even for obtaining large sample volumes, enabled

through cable car access and an ice tunnel dug along bedrock for touristic purposes; ii) previous work already demonstrated cold basal ice conditions, albeit more than 25 years ago (Haeberli et al., 1979; Lorrain and Haeberli, 1990); iii) in an ice cave, direct observations of the basal ice stratigraphy offer a more detailed picture of settings and potential processes than a small subsample obtained from an ice core. A short summary of the main site characteristics in given below.





## 2   Site characteristics and previous work

Glaciological investigations of the summit glacier at Chli Titlis started some thirty years ago in connection with the construction of a telecommunication tower (Haeberli et al., 1979). The cornice-type summit holds the glacier on its north-facing slope, the lee side. The summit itself features a tunnel through the bedrock connecting the cable-car station with the telecommunication

tower. Another tunnel was dug for tourist purposes around 100 m into the ice along bedrock starting at the cable car station. Based on their earlier reconnaissance, Haeberli et al. (2004) report sub-zero bedrock temperature as well as around $-1°$ C in the ice tunnel, with an increase towards its back end, with increasing ice/firn thickness and meltwater percolation through small crevasses at the end of the tunnel. The authors also report well-layered ice roughly 25 m thick and that accumulation rates on the flat summit area generally seem to be low. According to Haeberli et al. (2004), the existence of old ice at depth

appears likely due to negative temperatures reaching far into the underlying bedrock and basal flow velocity close to zero. From sampling ice at the ice/bedrock interface, Lorrain and Haeberli (1990) found a distinct shift towards more negative values in the stable water isotopologues ($\delta^{18}$O and $\delta$D). Although Lorrain and Haeberli (1990) doubt an Ice Age origin as the cause of this signal, the peculiar isotopic signature is nonetheless an important marker of the basal ice section, also found in other alpine glacier ice bodies (Wagenbach et al., 2012). Within the context of the study presented here, we reasoned that this basal

anomaly could only be preserved under cold and stagnant basal ice conditions, i.e. becoming temperate and/or enhanced ice flow would have likely erased this isotopic signature over the last 25 years. Accordingly, we made an attempt to re-find the basal isotope anomaly. Today's conditions at Chli Titlis are characterized by clear signs of a negative mass balance at the summit. The glacier section hosting the glacier cave is covered with fabric during the summer season to prevent further ice loss (which is evident through comparison with uncovered neighbouring glacier sections). The glacier has a remaining thickness

of 7–8 m above the ice cave (pers. comm. Christoph Bissig, Bergbahnen Titlis-Engelberg, 2017). Nowadays, active cooling is performed by ventilation of cold outside air into the ice tunnel, especially due to the cave being highly frequented by tourists.

## 3   Methods

A total of three campaigns have been conducted in January 2014, January 2015 and August 2015. Three profiles of ice blocks cut using a chainsaw were obtained from three different locations in the cave. Two profiles were cut out near the entrance of

the cave, aiming to be close to the original sampling location by Lorrain and Haeberli (1990). The third spot was located about 20 m deeper in the cave (Figure 1). All profiles were cut down to bedrock at locations where the ice/bedrock interface is clearly visible. Individual blocks were cut around 20 cm deep into the wall and varied, depending on the profile, between 10–20 cm and 8–17 cm in height and width, respectively (Figure 1). The deepest part of the profile 1 from 2014 showed a basal section of very clear, bubble free ice. This clear ice section at the base extended through a larger part of the cave, but was not present

at profiles 2 nor 3.





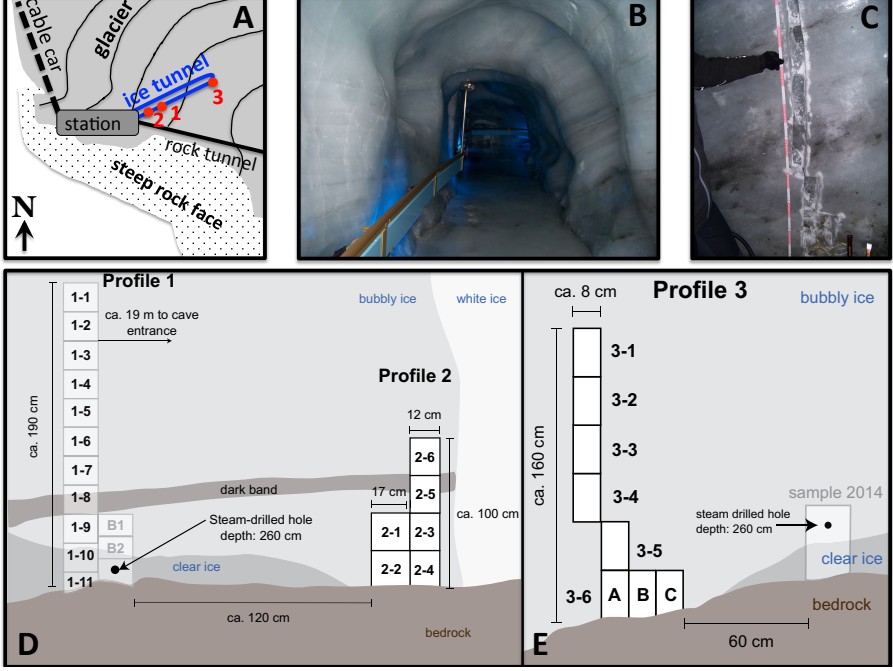

**Figure 1.** Overview on ice sampling at the Chli Titlis glacier. Figure part A shows a schematic diagram with ice sampling locations within the ice tunnel. Distinct near-horizontal layering is visible within the tunnel (B). A total of three ice block profiles was cut by chainsaw down to bedrock (C). Part D and E present schematic details on the ice block sampling. Note that two additional blocks (2-7 and 2-8) were cut in parallel behind block 2-4.

## 3.1 Englacial temperature

Only basic temperature measurements at about 10–15 cm depth in the ice wall could be performed during the initial reconnaissance campaign in January 2014. At the location of profile 1, a vertical profile of 9 holes was drilled for temperature measurements at 20 cm vertical distance. In the follow-up campaign in January 2015, two holes were drilled nearly horizontal by means of a stream drill (drilling slightly upward to let meltwater drain) just above bedrock roughly 2.6 m deep, at locations 1 and 3 (Figure 1). Temperature was measured by means of a negative temperature coefficient thermistor chain attached to a solid probe that was inserted to the boreholes. Calibration of the thermistors was done in a ice / water bath at $0°$ C. The quality of the temperature measurements is further determined by the data logger and the physical properties of the involved thermistors and cables. We estimate the maximum total accuracy to be $\pm0.2°$ C (cf. Hoelzle et al., 2011). Temperature readings were taken from the first thermistor at the end of the chain, i.e. at the deep end of the boreholes. To reduce latent heat effects that stem from drilling the borehole and to allow for the thermal adjustment of the thermistors, readings were taken at intervals of 4-10 minutes and were logged over 40–60 minutes. For all boreholes, this was sufficient to attain temperature fluctuations at least two orders of magnitude smaller than the estimated measurement accuracy.



**Table 1.** POC $^{14}$C dating results for the samples from Titlis glacier cave. The sample names denote the profile and the block number as indicated in Figure 1. The additional temperature in the sample name refers to the POC combustion temperature. The F$^{14}$C value is given according to the convention stated in Stuiver and Polach (1977). The calibrated ages have been calculated using OxCal version 2.4 (Ramsey, 1995). All calibrated ages are reported as their 1 sigma ranges in years before present (1950 C.E.).

| Sample name | POC mass [$\mu$gC] | F$^{14}$C | Calibrated age [yBP] |
|---|---|---|---|
| Titlis 1-2-800° C | 96.5 | $0.848 \pm 0.008$ | $1180 - 1305$ |
| Titlis 1-9-800° C | 47.7 | $0.702 \pm 0.007$ | $2861 - 3070$ |
| Titlis 2-3-340° C | 43.6 | $0.610 \pm 0.009$ | $4237 - 4615$ |
| Titlis 2-6-340° C | 20.1 | $0.754 \pm 0.009$ | $2122 - 2378$ |
| Titlis 3-5-800° C | 56.2 | $0.568 \pm 0.009$ | $5047 - 5319$ |

## 3.2 Stable water isotopes

All ice samples were stored in coolers with thermal packs and transported in frozen condition to the Institute of Environmental Physics, Heidelberg University (IUP-HD), for further analyses. The outermost 10 cm exposed to the tunnel were removed of each block. The opposite, i.e. inside-facing side of each block was used to obtain samples for stable water isotope analysis. Initially, each block was sampled at coarse resolution between 7-10 cm in distance along the vertical axis. To investigate lateral variability within the blocks, each coarse sample was further divided along its vertical axis (denoted samples A and B). The lowest 25 cm of profile 1 were cut at higher resolution (around 2 cm). Small uncertainty in assigning a distance above bedrock may stem from the slightly irregular block shape (round edges) and was estimated as accumulating to a few cm at most. All samples were analysed using conventional mass spectrometry ($\delta^{18}$O only) at IUP-HD. In Figure 2 the average of samples A and B is plotted against height above bedrock, using the absolute range in isotope values to indicate the lateral isotope variability. In order to obtain data for both water isotopes, $\delta^{18}$O and $\delta$D, additional measurements were performed using a Picarro cavity ring down spectrometer at IUP-HD. Co-isotopic measurements comprised all samples of profile 1 and, in addition, samples at high resolution (around 2 cm) of profile 2. However, $\delta$D values of profile 1 are associated with large uncertainty due to technical difficulties with the instrument at the time of measurement and hence have to be regarded with caution only.

## 3.3 Radiocarbon dating

For radiocarbon dating five different ice blocks have been analysed using the microscopic particulate organic fraction (POC). Two blocks each were selected of profiles 1 and 2, and an additional block of profile 3 (Table 1). The ice samples were melted, the POC was filtered, combusted into $CO_2$ and the radiocarbon content was measured via an accelerator mass spectrometer utilizing a gas ion source. Details on POC extraction and $^{14}$C measurement can be found in Hoffmann (2016). Visible dark layers and sediment contaminated parts were carefully avoided during sub-sampling. Upon processing for $^{14}$C analysis, all samples exhibited a thick layer of very black and highly organic material on the filter surface after filtration. Samples 2-3 and



**Table 2.** Overview on measurements of physical properties of ice samples from Chli Titlis. Indicated are sample and measurement type, i.e. using Large Area Scanning Macroscope (LASM) and Fabric Analyser (FA).

| Block | type of thin section | LASM | FA |
|-------|---------------------|------|-----|
| 1-9 | 2 vertical sections (6x7 cm) | yes | yes |
| 1-10 | 2 vertical sections (7x9 cm) | yes | yes |
| 1-11 | 1 vertical section, from clear ice (7x8 cm) | yes | no |
| 1-B1 | 1 vertical sections from the side (5x10 cm) | yes | yes |
| 3-2014 | horizontal section, just above clear ice (6x9 cm) | yes | yes |
| 2-7 | 2 sections in orthogonal planes: | yes | no |
|  | 1 vertical section from the back (7x10 cm), |  |  |
|  | 1 vertical section from the side (10x11 cm) |  |  |

2-6 have been combusted at $340°$ C. Samples 1-2, 1-9 and 3-5 were combusted at $800°$ C. Systematic investigations of samples with known [14]C ages revealed that the $340°$ C combustion temperature yields the best representation of the actual ice sample age. This effect was discovered during the course of this study but is discussed in detail elsewhere (Hoffmann, 2016). Due to different combustion temperatures of multiple organic species and increasing influences of aged and decomposed organic

5  material (reservoir effect), higher combustion temperatures can lead to higher [14]C ages. Thus, the retrieved ages for the $800°$ C combustion temperature samples are regarded as upper age limits.

### 3.4 Physical properties

To obtain complementary information about the physical properties, thick and thin section samples were prepared from four blocks of profile 1, from a sample of the clear basal ice in the back of the tunnel, and from one block of profile 2. The sections are

10  analysed using the Large Area Scanning Macroscope (LASM, Schaefter+Kirchhoff GmbH) to obtain microstructure maps of grain boundaries (GB) and bubbles, and with an automated Fabric Analyser (FA, Russell-Head Instruments), which provides the crystallographic orientation of individual ice crystals in a thin section sample. An overview of the samples is given in Table 2. The fabric analyser data was automatically processed (Eichler, 2013) and provides estimates for grain size and crystal-preferred orientation (CPO). The microstructure maps are qualitatively evaluated and discussed below.

15  ## 4 Results and Discussion

### 4.1 Cold basal ice conditions

The initial temperature measurements (in 2014) in shallow horizontal boreholes (mechanical drilling) showed little variability within the vertical profile, with $-2.3°$ C at about 1.8 m above bedrock, $-2.2°$ C just above bedrock and the average over




nine different measurements around $-2.3°$ C. Notably, even the shallow measurements differed unambiguously from the air temperature in the ice tunnel, which was measured at $-1.5°$ C. The subsequent measurements in the deeper horizontal boreholes (2.6 m, thermal drilling) at locations of the isotope profiles 2 and 3 revealed $-2.9°$ C and $-2.6°$ C respectively, in January 2015. For comparison, the temperature measurement was repeated in high summer, showing $-1.9°$ C at profile 2 in

August 2015, which is still substantially below zero. The results demonstrate that sub-zero englacial temperatures very likely prevail at our sampling sites in the ice tunnel, although the values should be considered as upper limit estimates. This is i) due to the limited equilibration time and latent heat effects from drilling of the borehole and ii) because of the higher air temperature in the ice tunnel that affects air temperature in the borehole and the temperature of the surrounding ice.

## 4.2 Basal isotope anomaly

As already stated above, the stable water isotope data primarily served as a general stratigraphic marker and, more importantly, was specifically investigated with respect to refinding the outstanding basal isotope signature. The isotope data of the three profiles are presented against height above bedrock in Figure 2. The average levels in $\delta^{18}$O are –13.35, –14.45 and –14.54 ‰ for profiles 1, 2 and 3, respectively, thus broadly consistent with the values reported by Lorrain and Haeberli (1990). Although the coarse resolution isotope levels are very similar among the upper parts, the profiles differ significantly regarding their basal

section. While no outstanding signal with respect to the rest of the profile is found in profile 3 (albeit available only at coarse resolution) the lowermost 10–20 cm of profile 1 and 2 comprise more enriched and depleted values, respectively. In profile 1, the lowermost 15 cm coincide with a basal layer visibly clear and free of bubbles (cf. Figure 1). The isotopic values of this section show little variability with respect to the rest of the profile, except for a stepwise increase by roughly 3 ‰ within the last 5 cm of the profile. Profile 2, on the other hand, features a gradual rise in $\delta^{18}$O values between 40 and 20 cm above bed,

followed by a sharp drop by about 2 ‰ within the lower 20–15 cm. The differences in basal isotope signature between profile 1 and 2 are also reflected in different microstructural properties (see below). With values around $-16$‰ just before bedrock (clearly outside of the range of the rest of the profile), the basal isotope signature of profile 2 is in near-perfect agreement with the phenomenon described by Lorrain and Haeberli (1990). Sampling an adjacent block of roughly the same depth at the location of profile 2 reproduces this anomaly, confirming its presence at this location (cf. grey line in Figure 2). A basal

isotope signature characterized by a gradual enrichment followed by distinct depletion in isotope values has been also observed at various other mountain ice core drilling sites, in particular at Colle Gnifetti and Mont Blanc (Wagenbach et al., 2012) which may suggest a common underlying mechanism (Keck, 2001). An adequate investigation of this intriguing phenomenon remains outside the scope of the present work. However, the re-discovery of the isotope anomaly initially described at Chli Titlis by Lorrain and Haeberli (1990) is significant in the following sense: Its unchanged presence since the last 25 years strongly

indicates that the basal ice has not undergone substantial changes as would be expected if having become temperate or under significant basal ice movement.

Using the $\delta$D values available for profile 1 and 2 we also performed a co-isotopic analysis for further comparison between the two profiles. Calculating linear regression of $\delta$D against $\delta^{18}$O yields slopes of $(7.7 \pm 0.2)$ and $(7.6 \pm 1.0)$ for profile 1 and 2, respectively (reported with one standard error). These values are near-identical to the previous investigation (Lorrain and



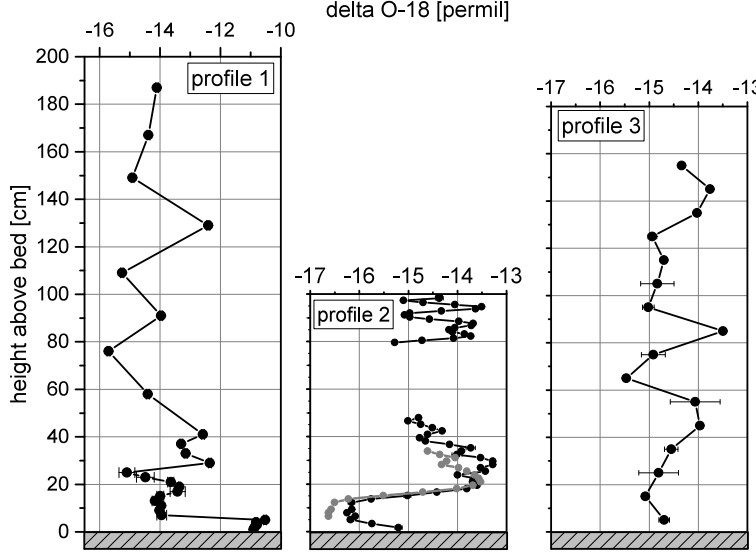

**Figure 2.** Stable water isotope profile of the three sampling locations. Profile 2 reproduces the basal anomaly previously described by Lorrain and Haeberli (1990). The grey plot shows measurements of a neighboring block. The data gap in profile 2 corresponds to block 2-5, for which isotope measurements are not available. Note that the comparatively less negative isotope values close to bedrock in profile 1 correspond to the layers with clear ice (see text). No distinct basal isotope was found in profile 3. Error bars denote the range in values from two adjacent samples that were analyzed and the results averaged (profile 1 and 3).

Haeberli, 1990). We note a greater degree of scatter among $\delta$D and $\delta^{18}$O in profile 1, reflected in the larger standard error of the slope and smaller correlation coefficient with respect to profile 2 (i.e. $r = 0.84$ vs. $r = 0.99$ for profile 1 and 2, respectively). However, $\delta$D values of profile 1 are regarded with caution only and hence were not interpreted at more detail.

### 4.3 Ice microstructure

All microstructural samples were taken from the lower parts in both profiles. The mean grain size, derived from the FA images, lies between 0.55 and 2.4 cm$^2$ but the largest grains cover between 10 and 20 cm$^2$. The grains show little indication of active deformation (e.g. no abundant subgrain boundaries) and it can be assumed that the large grains are a consequence of normal grain growth, which is the dominant microstructural process in stagnant ice and strongly enhanced by warm temperatures (Faria et al., 2014). While this finding holds for the entire basal section of profile 2, including the lowermost 10 cm, the lowermost 10 cm of profile 1 (block 11) include many smaller grains and are characterized by a laminar or elongated grain structure with irregular grain boundaries and almost bubble-free, corresponding to the basal sections identified as clear ice (Figure 3). Accordingly, the basal microstructural pattern of profile 1 independently confirms the presence of ice from refrozen meltwater. Additionally, in the basal sample of profile 2 that was cut perpendicular to the tunnel wall, elongated bubbles (mean aspect ratio





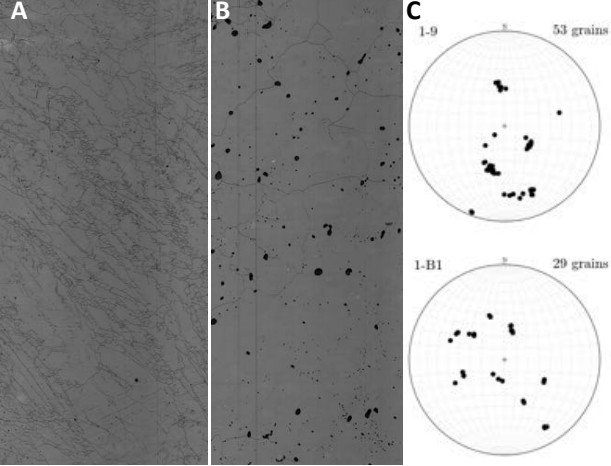

**Figure 3.** Exemplary results from microstructural analysis. Two exemplary images of a Large Area Scanning Macroscope (LASM) of basal ice in profiles 1 and 2 are shown in A and B, respectively. The basal ice of the two profiles looks distinctly different, with large grains and abundant air bubbles dominating profile 2 (B). In contrast, the lowest 10 cm of profile 1 are almost bubble-free and show very small elongated grains (A), indicating refrozen meltwater (see text). Schmidt diagrams shown in C illustrate the multiple maxima crystal-preferred orientation of the (non-meltwater) basal ice.

of 1.8) inclined at approximately $45°$ were observed. This implies that a moderate deformation is to be expected, either due to the lacking constraint of the tunnel and/or the decrease in viscosity due to the warm temperatures. The CPO pattern observed in the large-grained samples can be described as a multiple maxima pattern similar to the CPO of Eemian ice (Dahl-Jensen et al., 2013).

## 4.4 Cold basal ice at Chli Titlis

Based on our evidence from englacial temperature measurements revealing sub-zero conditions, a preserved basal isotope anomaly and analysis of physical properties, we conclude that almost stagnant ice frozen to bedrock still exists at Chli Titlis today. This is not to be expected a-priori in view of ongoing warming conditions. For instance, a temperature of only $-0.7°$ C at 15 m depth was reported in 1979/80 by Haeberli et al. (1979) at the summit firn of Chli Titlis, and a temperature inside the ice tunnel of around $-1°$ C reported in 1990 (Lorrain and Haeberli, 1990). At the same time, atmospheric warming trends of the past decades are reflected in rising englacial temperatures even at the highest glaciers above 4000 m asl in the Western Alps (Gilbert and Vincent, 2013; Hoelzle et al., 2011). Titlis glacier is reported to show a negative mass balance for the time period 1986-2010 (Zemp et al., 2015, doi: 10.5904/wgms-fog-2017-06), consistent with evidence for negative mean geodetic mass balance of Swiss glaciers between 1980–2010, extending to locations above 3500 m asl (Fischer et al., 2015). No direct mass balance measurements with stakes close to the ice cave have been carried out. The glacier wide mass balance data provides only limited information regarding the mass balance at the ice cave, but draws a general picture. A comparison to other glaciers at similar elevations would have to take into account the local climatological settings (Abermann et al., 2011) and, in addition,



stake measurements in summit locations are generally sparse. In the Eastern Alps, mass balance is measured for the full altitude range with stakes on Kesselwandferner in Oetztal Alps. The stake L8, located at fairly the same altitude in a similar climatic setting as Chli Titlis changed from being close to ELA to increasing mass losses since the mid 1980s (Fischer et al., 2014). The above evidence suggests that the ongoing change may have also affected the lowermost ice sections at Chli Titlis over the past

decades, although making a straightforward connection between atmospheric warming trends and conditions in the ice cave suffers from a great deal of complexity. Investigation of ice masses in carst caves pointed out that mass and energy balance in cave systems are more intricate (Schöner et al., 2011; Obleitner and Spötl, 2011) than for a glacier without englacial cavities as it would be the case when drilling an ice core. For instance, this potential warming influence is counteracted by ongoing efforts to actively control the air temperature in the ice tunnel and to protect the surface from ablation, which may contribute

to keeping the ice frozen to bedrock. Surface covers reduce ablation (Fischer et al., 2016), and alter the surface energy balance mainly by reducing the direct incoming solar radiation (Olefs and Lehning, 2010). The propagation of these changes in energy balance into the glacier, and potential changes in the ice temperatures have only been investigated in a depth of 3 m ice on Schaufelferner in the Stubai Alps with temperatures close to $0°$ C during summer. No significant differences between minimum ice temperatures of covered and uncovered areas had been evident then. Trying to disentangle these anthropogenic technical

measures from natural effects is difficult. However, the above considerations illustrate the complexity of the situation, while raising doubts as to what extent the cold basal ice conditions would have been preserved without the current technical measures. Predicting the future fate of Chli Titlis glacier would certainly require a separate dedicated investigation.

### 4.5 Constraining the maximum age of Chli Titlis ice

Basal ice frozen to bedrock is the fundamental prerequisite for the ice to reach substantial age at small glaciers like Chli

Titlis. The results of the radiocarbon dating efforts reveal samples with generally very large POC concentrations starting at 500 $\mu$gC/kg of up to almost 4 mgC/kg ice. This includes samples without any visible inclusions of particulate material. The POC concentrations are a factor of 10–100 higher than for other high alpine glaciers (e.g. at Colle Gnifetti (Hoffmann, 2016)). These high concentrations result in relatively small statistical $^{14}$C counting errors of $1-2\%$ and also no significant influences of process blanks are expected. All radiocarbon ages have been calibrated using OxCal 4.2 (Ramsey, 1995). Figure 4 presents

the calibrated ages reported with their 1-sigma range, and gives an overview of the location of the dated ice blocks of profiles 1 and 2. We generally find younger ages at greater distance above bedrock in each profile. This suggests a general chronological order of the layers in the ice tunnel wall, which is also expected considering its distinct visible layering free of folds or other stratigraphic disturbances. It should be noted, however, that the layering appears to be not strictly horizontal, but slightly tilted towards the back end of the tunnel. In particular, we find a distinct dark band close to bedrock and running through profiles 1

and 2. Tracing the sharp lower edge of the dark band leads from about 63 cm at profile 2 (roughly at the center of block 2-5) to about 41 cm at profile 1 (incidentally right at the border between blocks 1-9 and 1-8). It remains difficult from our data to precisely identify the physical cause of the dark band, e.g. if this layer could be indicating a former hiatus in glacier growth. However, tracing the dark band between profile 1 and 2 suggests that at least parts of the lower 63 cm of profile 2 are not present at profile 1. Having found evidence of refrozen meltwater at the base of profile 1 suggests basal melting has potentially





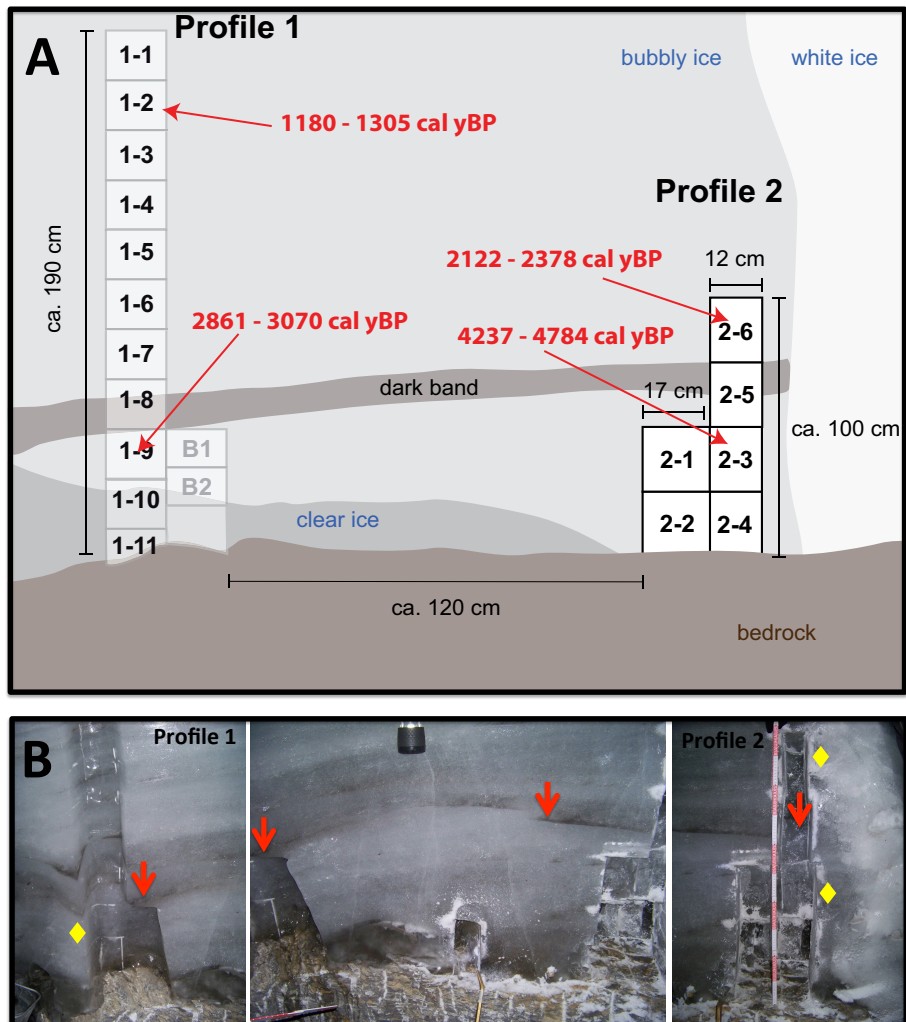

**Figure 4.** Calibrated radiocarbon ages of profiles 1 and 2 (in red, part A). Bottom row, part B: Collection of three pictures taken during the 2015 sampling campaign to illustrate the visual layering between profiles 1 and 2. The lower end of the visible dark band (indicated by red arrows) is located at about 41 and 63 cm above bedrock for profile 1 and 2, respectively. Yellow diamonds indicate the location of radiocarbon dated samples (see text).

played a role in causing this deficit. In this view, it appears reasonable to find block 1-9 at least about 500 years older than block 2-6 but more than 1000 years younger than block 2-3 (at a similar distance above bedrock). However, sample 1-9 was combusted at $800°$ C, which may partially contribute to slightly higher $^{14}$C ages compared to the samples combusted at $340°$ C (e.g. 2-3 and 2-6). Worth noting in this context, the maximum ages found at profiles 2 and 3 (both without the bubble-free basal layer) differ only by about 450 years, based on combustion temperatures of $340°$ C and $800°$ C, respectively. We took great care in sub-sampling the ice blocks to avoid any age biasing contaminants such as old cryoconite or organic sediment.





For this reason, the lowest and arguably the oldest samples (roughly the lowest 10-20 cm) have not been analyzed for $^{14}$C, since showing abundant inclusions of small rocks and sediment (although these samples may serve for a future dedicated investigation into age-biasing processes in radiocarbon glacier dating). Taking all this into consideration, the oldest basal sample (5047–5319 years BP, profile 3) is regarded as representing an upper age limit for ice at Chli Titlis.

The large gradient in age (e.g. for profile 2, several 1000 years difference within less than a 1 m) also deserves attention. Numerical flow modelling by Haeberli et al. (1979) has indicated the presence of substantial shear near bedrock even close to the summit, making thinning of the lowermost layers a possibility. From our analyses we also find evidence of basal deformation in the ice microstructure. However, with congelation ice as the main process of ice formation at the site, phases of ablation (like observed today) followed by recurring accumulation could also play a role in explaining the observed large vertical gradient

in age. Reconstructing the details of the glacier response to past (and future) climate conditions deserves a separate thorough investigation, e.g. taking into account numerical modelling of ice flow and energy balance. As a potential contribution to such an effort, the results of the study presented here already constrain the maximum age of the ice remaining at the summit of Chli Titlis today.

## 5   Outlook

Based on the above results, promising future contributions can be expected from extending the approach presented here in a more systematic investigation to constrain the age of other summit glaciers, selected with respect to comprising cold-based ice condition, access to basal ice and geographic coverage (preferably also in the Eastern Alps). Taking as a promising example in this regard, we have also started to investigate basal ice of the Schladminger glacier at Hoher Dachstein (Austria). This site offers similar glaciological and ice sampling conditions like Chli Titlis: Schladminger glacier is located at 2700 m asl on a

cornice-type, north facing summit, and direct access to basal ice becomes possible by means of an ice tunnel for tourists. We have employed the same set of tools described here, and found an englacial temperature of $-1.3°$ C at 190 cm inside the tunnel wall just above bedrock. As a preliminary result, radiocarbon analysis of one of the obtained basal ice blocks indicates an age around 4297–3715 yBP (1 sigma range) which is, interestingly, close to the age range observed at Chli Titlis.

The maximum age constraints of the basal ice at Chli Titlis suggest that this location has been ice-covered over roughly the last

5000 years. This finding is in general agreement with widespread evidence of a period of minimal glacier extent throughout the Alps at that time (e.g. Ivy-Ochs et al., 2009; Hormes et al., 2001). This includes evidence from dating archaeological artefacts recovered at other summit locations of comparable altitude, such as possible ice-free conditions at Tisenjoch (3210 m asl, the location of the Oetztal ice man) (Bonani et al., 1994) and, at greater proximity to Titlis, Schnidejoch pass (2730 m asl) (Grosjean et al., 2007). In order to investigate if the evidence from Titlis and Schladminger glacier point in fact towards a

widespread phenomenon of reduced summit glaciers in the time period around 4000-5000 years BP, it will be intriguing to continue our analysis and to integrate additional Alpine summit glaciers at comparable altitude.

## 6 Conclusions

We have successfully employed a combination of englacial temperature measurements with ice analysis and radiocarbon dating to show that ice frozen to bedrock still exists at Chli Titlis, and were able to constrain the maximum age of the ice remaining at the summit today. For this purpose we utilized an existing ice cave to directly access, sample and investigate the age,

isotopic and physical properties of the basal ice layers. Englacial temperature measurements show sub-zero ice temperatures throughout the year, albeit likely influenced by the air temperature in the tunnel. This finding indicates close-to-stagnant ice frozen to bedrock, substantiated by results from ice microstructure analysis. In addition, the stable water isotope measurements obtained from one profile reproduces a particular basal anomaly found in a study performing the first ice sampling over 25 years ago. Our radiocarbon analysis of five ice blocks suggests an chronological order of the visible ice layers and gives a constraint

of the maximum age of the lowermost sections of maximal 5000 years before present. The latter result is consistent with existing evidence suggesting for that time ice-free conditions at summit sites of comparable altitude. Based on the success of our approach we have already extended the investigation to similar sites in the Eastern Alps, with promising first from an ice tunnel at Schladminger glacier. The results of the study presented here demonstrate that, even today, cold-based ice still persists at summits of substantially lower altitude than 4000 m asl. For Chli Titlis, although mass loss is ongoing today even at

the summit, this location has likely been ice covered for at least the last 5000 years.

*Competing interests.*   The authors declare that they have no conflicts of interest.

*Acknowledgements.*   We gratefully acknowledge the support of the Bergbahnen Titlis-Engelberg, in particular the help of Christoph Bissig and Peter Reinle. Likewise we acknowledge the support of the Dachstein Gletscherbahn Ramsau. Part of this work was performed under the project "Cold Ice" funded by the Austrian Science Fund (FWF): P 29256-N36. Financial support was provided to P.B. by the Deutsche

Forschungsgemeinschaft (BO 4246/1-1). J.K. received funding by the Studienstiftung des deutschen Volkes and support by the Helmholtz Junior Research group VH-NG-802. The Klaus-Tschira-Lab Mannheim is acknowledged for their support in radiocarbon analysis. We also thank Martina Schmidt and Michael Sabasch of the Institute of Environmental Physics, Heidelberg University for their support with the stable water isotope analysis. We would like to especially thank and acknowledge our late colleague Dietmar Wagenbach, who had initiated our work at Chli Titlis and contributed his long-standing experience at the beginning of the project.



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
