# Peer review of "Investigating cold based summit glaciers through direct access to basal ice: A case study constraining the maximum age of Chli Titlis glacier, Switzerland"

_The Cryosphere, 2017_

## Referee Comment (RC1) · B. Laabs (Referee) · 2 Nov 2017

The manuscript describes a useful and important study of basal-ice in Chli Titlis glacier. The methods are described well and represent a preliminary investigation that the authors have already extended to other such glaciers. The manuscript is interesting and suitable for The Cryosphere. I believe it can be improved by addressing the comments provided below, which would constitute a minor revision.

Two items of importance are summarized here; the remaining comments are minor.

[Figure]

1. Please provide a location map of the Chli Titlis glacier and the neighboring glaciers described in the study. This will be useful for the readership that is less familiar with the study area and the regional glaciers.

2. The authors seem to waiver on the reliability of the del-D data for basal ice, citing some concerns about the quality of the measurement. The data are described as a regression with del18O, but are not illustrated. The authors should decide whether to include the data; based on the information given in version 1, the inclusion of the del-D data may not be necessary.

Minor comments:

Page 1, Line 16: the term "sedimentary glaciers" is unconventional. Consider using a different term to characterize such glaciers. Perhaps you mean "stratified glacier ice"?

Page 1, Line 19: including an altitude range for "uppermost summit" glaciers?

Page 2, Line 19: change "arrive just-in-time" to "are now available"

Page 2, Line 23: change "offers constraining" to "limits"

Page 2, Line 33: final sentence of paragraph is not necessary.

Top of Page 3: a location map of the study area and the glacier would be useful here.

Page 3, Line 7: reword to clarify meaning of the phrase, "with an increase towards its back end".

Page 3, Line 17: insert sentence describing the "clear signs" of negative mass balance

Page 4, Line 7: change "a" to "an"

Page 5, Line 10: change "that stem from" to "caused by"

Top of Page 5: seems that radiocarbon dates appear too early. Move this table to a position after the "Radiocarbon dating" section.

Page 5, Line 13: the ïĄďD data are not shown in Figure 2 and are perhaps not necessary to discuss here.

Page 6, Line 1: seems that a reference for the "systematic investigations" should be provided here, or described further if they were done in this study. In either case, please provide more information beyond that given at line 3.

Page 7, Line 32: here the ïĄďD data are implied to be more reliable than stated on page 5 (in Methods). Some statement of potential error in the ïĄďD should be included here, as should a figure showing the regression of the ïĄďD and ïĄď18O data.

Figure 2 caption: the statement "no distinct basal isotope was found in profile 3" is unclear. Reword the statement to clarify it.

Page 9, Line 2: define CPO and expand on the last sentence to explain its significance.

Page 10, Lines 1-5: here, the reader would benefit greatly from a location map showing the study area relative to the nearby glaciers with mass balance measurements.

---

## Referee Comment (RC2) · R. Waller (Referee) · 28 Nov 2017

General comments

I agree with the first referee that this paper presents a series of interesting findings from a summit glacier in Switzerland that suggest that a cold-based thermal regime has been persistent at this site resulting in the preservation of the basal ice for c. 5,000 years. The paper is therefore clearly appropriate for publication within The Cyrosphere although I think there are areas that would benefit from further work.

[Figure]

Areas for improvement

â Ăć Agree firstly with the comments of the first referee - particularly the need to include a map of the study site that indicates the location and setting of the glacier and the tunnel. â Ăć The paper focuses on the examination of basal ice at the site but is unclear whether the use of the term "basal" is used simply to refer to its position at the base of the glacier or in its glaciological sense (ice which is produced at and interacts with the bed; e.g. Knight, 1997). Either way, I would recommend that the authors consult some of the relevant literature to inform their description and interpretation of the ice examined in this study. â Ăć On a related note, I would like to see a more detailed description of the ice facies observed within the tunnel within section 3 to support the more detailed ice petrography reported in section 3.4 (see for example Hubbard & Sharp, 1995). â Ăć The impact of the work would be enhanced if greater emphasis was given to the broader context of the work and its key findings within both sections 1 and 5. What is the wider palaeoclimatological and palaeoglaciological significance of the preservation of ice 5,000 years at this altitude? â Ăć Section 4.5 in particular would benefit from a clearer structure to help emphasise and explain the key points.

Minor comments P1 - Abstract - Highlight the primary research question this research is aiming to address. Feel this will help to establish its wider context and significance. P1 - L8/9 - Explain what is meant my "standard glaciological tools". P1 - L9 - Clarify what is meant by the use of the term "sophisticated". P1 - Section 1 - Explain more explicitly why cold-based thermal conditions are of such importance - i.e. warm-based conditions and basal melting lead to the loss of the oldest ice - impossible therefore to date onset of most recent phase of glaciation. P1 - L19/20 - Provide the approximate altitudinal ranges for "uppermost summit ranges" and "lower altitudes". P2 - L4 - Basal temperatures persistently below the pressure melting point? P2 - L5 - Clarify what is meant by "glacier buried tree parts" - re-word. P2 - L8/9 - Give greater emphasis to this key broader aim of the research (e.g. could be presented at the start of the final paragraph in this section) and provide a little more explanation on how the paper will help

to realise this aim. P3 - L6 - "as well at around..." P3 - L8 - What attribute provides the layering? Variations in bubble content, sediment concentration? As mentioned earlier, providing a more detailed description of the characteristics of the basal ice here and within section 4 would be helpful. P3 - L25 - Reword from "third spot" to "third profile". P3 - L28/29 - Use of the term "clear" here needs further clarification. Again - highlights need to include a section (maybe initially in section 2) that provides a more detailed description of the basal ice facies observed and clarification of the significance of the use of the term "basal". P4 - Figure 1 - Include scale in Figure 1A. P4 - L4 - "20cm vertical intervals" P5 - Section 3.2 - Where the stable isotope measurements taken from all the ice blocks? (Fig 2 suggests not) P5 - L3 - "The outermost 10 cm of each block exposed to the tunnel was removed" P5 - Section 3.3 - Which blocks were used for the radiocarbon dating? P6 - Section 3.4 - Include a description of the macroscopic characteristics of the ice facies investigated here - ideally refer to an ice facies classification scheme. Explain why the clear ice facies was specifically targeted for analysis. P6 - Section 4.1 - It's worth emphasising here that the measured temperatures are significantly lower than those previously recorded by Haeberli. P7 - L7 - Equilibriation? P7 - Section 4.2 - Explain the significance of a replication of the basal isotope anomaly. Does this indicate that the basal ice formed from precipitation during colder climatic conditions? P8 - Figure 2 - Illustrate which samples have been obtained from the clear ice (cf. Figure 1D/E). P8 - Again, a brief description of the ice facies and their key characteristics (e.g. debris content and bubble content) would help provide a context for the microstructural characteristics. P9 - Figure 3 - Where have have these results been obtained from? "Selected results" rather than "exemplary results". P9 - Section 4.4 - Does progressive downwasting and thinning of the ice provide a potential explanation for the fall in temperature? P9 - L6 - Suggest rewording to - "...from basal temperature measurements revealing temperatures well below the pressure melting point" P10 - L6 - "karst" rather than "carst" P10 - L19/20 - Emphasise why a cold-based thermal regime is required to preserve old basal ice. P10 L19-22 - What is the source of this organic material and how has it been incorporated into the basal ice? P10 - L26-28 - What is the layering composed of? Variations in bubble content and ice crystal size? Suggestion here would be that this could be foliated ice in which case it could be illustrative of shear deformation. P10 - L29 - What is meant by the use of the term "dark". Bubble free? P12 - L6-7 - Earlier text suggested limited deformation whilst this suggests the potential for deformation - watch for possible contradictions here. Would typically expect significant shear strains in the basal ice layers of non-temperate glaciers frozen to hard rock beds. P12 - L24-25 - Key finding - Give greater emphasis within the section. P13 - L5 - Suggest re-wording to: "Tmperature measurements demonstrate basal temperatures that are well below the pressure melting point..." P13 - L9 - "...five ice blocks suggests a chronological order..." P13 - L14-15 - This final sentence is the key finding. Give greater emphasis (new paragraph?) and elaborate briefly on the potential palaeoclimatological and palaeoglaciological implications.

References Hubbard, B. & Sharp, M., 1995. Basal ice facies and their formation in the Western Alps. Arctic & Alpine Research, 27(4), 301-310. Knight, P.G., 1997. The basal ice layer of glaciers and ice sheets. Quaternary Science Reviews, 16(9), 975-993.

———————————

---

## Referee Comment (RC3) · B. Hubbard (Referee) · 4 Dec 2017

General comments

The paper reports ice temperatures, stable-isotope concentrations and radiocarbon dates from ice located near the base of Chli Titlis, Switzerland. The manuscript reports that the basal ice is cold (and infers that this has been the case for a considerable period of time) and that it has a maximum age of ∼5000 years BP. I find the manuscript to be of interest, but also that it still needs fairly substantial revision and improvement

in a few areas. My main concerns relate to two aspects of the manuscript.

1. The measured ice temperatures at the location are too close to the glacier's surface and/or the access tunnel to be able to ignore these external influences. Seasonal variations in surface temperature typically penetrate 10 to 15 m into the ice. Further, temperature within the tunnel is artificially influenced and, to some extent, controlled (as clear from p. 7 paragraph 1). The reported temperatures are also not substantially below freezing, so error 'estimated at 0.2 degrees C' really needs a more formal quantitative analysis and accompanying statement. I recommend plotting the thermistor time series to reflect ice cooling back down (presumably exponentially) to host temperature following borehole-wall warming by the steam drilling. I believe this aspect of the manuscript needs greater focus and for all the relevant information to be brought into one early section. For example, published temperatures are reported towards the base of page 9 that should really form part of this background material. Temperature control in the tunnel is also mentioned elsewhere later in the manuscript.

2. The manuscript reports on the temperature of basal ice, but presents no formal analysis of the nature that ice. Such an analysis would be useful both for the reader to understand the nature of the environment, and because certain physical basal-ice facies are indicative of certain basal processes and conditions. I recommend the revised manuscript include a formal analysis of the types of ice present at the sample locations (see review by Hubbard and others (2009) in Quaternary Science Reviews and references therein).

Specific comments (page/line):

1/1 "with great success" adds little and is a value judgement; I recommend deletion.

1/3 "... low altitudes may also contain old ice if locally frozen..."

1/5 "However, with recent warming and consequent glacier mass loss,..."

1/6 Delete ", however," and "Since sampling and dating the lowermost ice usually
requires. . ."

1/8 "We combine standard glaciological tools. . ." (and such 'tools' should just be specified as the term is rather too open)

1/9 ". . . physical properties and radiocarbon dating."

1/11 this "pioneering exploration" needs to be specified for the definite article ("the") replaced. (Note: I have not gone through the rest of the manuscript in the same detail; the grammar can still be improved)

2/15 These statements make clear the need for a formal analysis of the ice types present and sampled for this study.

2/23 What is "glaciological surveying"?

3/9 ". . . generally seem to be low" ideally needs some specification and quantification.

3/18 - 21 I recommend combining this material with temperature data from the existing literature in order to present as complete and accurate a situation as possible relating to the thermal history of this site. That this ice is, and has been, cold is central to the manuscript's message.

5/8-14 I think the manuscript would benefit from a more formal statement of isotopic error here. Currently, some delta D values are issued with caution because of 'large uncertainty'. I would prefer to see formal error bars added to each data point.

7/25 Tes, there appears to be a pattern here that broadly matches one(s) recorded elsewhere, but such a comparison should include all other profiles (including ice coring literature from Alpine glaciers at least) so the reader is convinced that this particular pattern is significantly over-represented. Also, if it is real, the explanation is a little truncated. Could it be related to the formation of clear facies basal ice by deformation-induced preferential expulsion of light isotopes?

7/33-35 I'd like to see this co-isotopic plot (including error bars). Which of the less

certain delta D values were used and what is their associated error. If the data are not of sufficient quality to 'interpret in more detail' then they may not be of sufficient quality to present at all; at present, the reader cannot judge this.

8/5-8 Is it possible to illustrate these crystal size differences and the sub-grain boundary and elongation conditions mentioned? The elongated crystals sound like 'interfacial' facies ice, agreeing with the congelation origin advanced in the manuscript. The text states 'grain size' – which is presumably 'ice crystal size'.

9/Fig. 3 These images are not very clear and seem to give no indication of scale

9/2-4 This reference to possible warm temperatures in the past seems at odds with the general thermal interpretation of the site as cold. Perhaps some text could be spent on rationalizing these seemingly contrasting thermal conditions.

10/6 'karst'

10/9-10 More here of relevance to the thermal conditions

10/17 True, but to focus on the future does not address the issue raised by this paragraph – that there may be issues complicating the temperatures reported... Surface temperature changes should be accounted for in any interpretation of point temperatures recorded within a thickness of $\sim 10$m. Same for the tunnel, although this zone of influence is likely smaller because the temperature changes in the tunnel will presumably be muted. Are there records of external temperature at the surface (or nearby, to which a lapse rate can be added) and in the tunnel?

11/Fig. 4 Some of the structures here do seem to indicate ice deformation. How does this relate to the interpretation of generally undeformed ice in this location? Maybe these features are not deformation structures, but some analysis and interpretation might help address this possible issue.

---

## Author Comment (AC1) · 19 Dec 2017

The comment was uploaded in the form of a supplement:
https://www.the-cryosphere-discuss.net/tc-2017-171/tc-2017-171-AC1-supplement.pdf

---

## Author Comment (AC2) · 19 Dec 2017

**"Investigating cold based summit glaciers through direct access to basal ice: A case**

**study constraining the maximum age of Chli Titlis glacier, Switzerland" by Pascal**

**Bohleber et al.**

- Response to reviews -

*Please note:*

- *Author's responses to the referee's comments are in blue*

- *Changes in the corresponding revised manuscript are highlighted in red*

- *All line numbers in "Changes to manuscript" refer to the new revised version*

- *All new references can be found in the new manuscript*

**Response to referee #2, R. Waller**

General comments

I agree with the first referee that this paper presents a series of interesting findings
from a summit glacier in Switzerland that suggest that a cold-based thermal regime
has been persistent at this site resulting in the preservation of the basal ice for c.
5,000 years. The paper is therefore clearly appropriate for publication within The
Cyrosphere although I think there are areas that would benefit from further work.

We thank the referee for his comments and in particular for bringing our
attention to the importance of a detailed description of the ice facies. Although
the authors did not have a detailed background in geology we have consulted
the respective literature and our visual analysis of the stratigraphy. Based on
this further investigation we have added to the revised manuscript, in
particular Table 2 giving an overview on the macroscopic ice characteristics.

Areas for improvement
Agree firstly with the comments of the first referee - particularly the need to include
a map of the study site that indicates the location and setting of the glacier and the
tunnel. The paper focuses on the examination of basal ice at the site but is unclear
whether the use of the term "basal" is used simply to refer to its position at the base
of the glacier or in its glaciological sense (ice which is produced at and interacts
with the bed; e.g. Knight, 1997). Either way, I would recommend that the authors
consult some of the relevant literature to inform their description and
interpretation of the ice examined in this study. On a related note, I would like to see
a more detailed description of the ice facies observed within the tunnel within
section 3 to support the more detailed ice petrography reported in section 3.4 (see
for example Hubbard & Sharp, 1995). The impact of the work would be enhanced if
greater emphasis was given to the broader context of the work and its key findings within both sections 1 and 5. What is the wider palaeoclimatological and
palaeoglaciological significance of the preservation of ice 5,000 years at this
altitude? Section 4.5 in particular would benefit from a clearer structure to help
emphasise and explain the key points.
We appreciate these valuable suggestions and have tried to integrate them in
the revised manuscript. Specifically, we have compiled a new figure 1 showing
a location map. We have also consulted the suggested literature regarding ice
facies, and now include these references in a more in-depth discussion of this
point. We have added a respective table (Table 2) with an overview of the
main stratigraphic features at each sampling site. We also tried to add more
discussion regarding the wider paleoclimatological and paleoglaciological
significance of our results. At the same time we would like to point out the
pilot character of this work and believe that the full paleoclimatic implications
of constraining the maximum age of these summit glaciers will fully unfold
after combining results from several sites and comparing them with other
proxy evidence in more detail. This is already part of our ongoing
investigation.
As an additional comment, we now realize that the use of the word "basal" in
the original version of the manuscript was not sufficiently clear and may have
caused some confusion. We intended to refer to the lowermost, hence
potentially oldest sections of the glacier. While this includes the "basal layer"
in a strict glaciological sense, we actually referred to a much larger section of
ice above the glacier base, i.e. that becomes accessible at Chli Titlis through the
ice cave.
We thank the referee for pointing out this ambiguity and have clarified the
manuscript accordingly.
**Changes to manuscript:** P4, L4-5: Clarified the use of the term "basal" vs
"lowermost".
Minor comments
P1 - Abstract - Highlight the primary research question this research
is aiming to address. Feel this will help to establish its wider context and
significance.
Thank you for the suggestion. We decided to reword part of the abstract to
point out our primary research questions more clearly.
**Changes to manuscript:** P1, L6ff: Reworded abstract.
P1 - L8/9 - Explain what is meant my "standard glaciological tools".

We have removed this part in the reworded abstract version and feel that our
approach is presented in a more explicit way now, including specifically
stating what tools were used in the study.
P1 - L9 - Clarify what is meant by the use of the term "sophisticated".
We have reworded this part as "state-of-the-art micro-radiocarbon analysis".
The primary challenge compared to conventional radiocarbon dating methods
is that glacier ice comprises extremely low carbon concentrations, requiring a
great deal of sophistication in sample preparation and analysis.
**Changes to manuscript:** P2, L22ff. We have clarified this point in the Introduction
and also include a new reference to the recent paper by Hoffmann et al. (2017),
describing the employed technique in detail.
P1 - Section 1 - Explain more explicitly why cold-based thermal conditions are of
such importance - i.e. warm-based conditions and basal melting lead to the loss of
the oldest ice - impossible therefore to date onset of most recent phase of glaciation.
Thank you for the suggestion. We have added a respective statement to clarify.
**Changes to manuscript:** P2, L6-8. Statement added.
P1 - L19/20 - Provide the approximate altitudinal ranges for "uppermost summit
ranges" and "lower altitudes".
Changed accordingly.
P2 - L4 - Basal temperatures persistently below the pressure melting point?
Yes, thank you- changed accordingly.
P2 - L5 - Clarify what is meant by "glacier buried tree parts" - re-word.
Changed accordingly.
**Changes to manuscript:** P2, L12-13: " trees formely buried by glacier advances"
P2 - L8/9 - Give greater emphasis to this key broader aim of the research (e.g. could
be presented at the start of the final paragraph in this section) and provide a little
more explanation on how the paper will help to realise this aim.
We appreciate the suggestion and, in an attempt to give greater emphasis on
the broader context of this research, have restructured the middle part of the
introduction.

**Changes to manuscript:** P2, L5-29. Restructured part of Introduction.

P3 - L6 - "as well at around..."

> What we intend to say here is " report sub-zero bedrock temperature *and* temperatures around -1 deg C..."

**Changes to manuscript:** P4, L18-19. Changed statement.

P3 - L8 - What attribute provides the layering? Variations in bubble content, sediment concentration? As mentioned earlier, providing a more detailed description of the characteristics of the basal ice here and within section 4 would be helpful.

> At this instance we are referring to the earlier study by Haeberli et al. (2004). The authors do not mention any details regarding the nature of the layering. However, following the referee's suggestion we have added more detail regarding the stratigraphy of the three sampling sites. We have included this description in section 3.4 (see comment below), and have also added a new table (Table 2) summarizing the main characteristics. We then refer to these characteristics again in section 4.

**Changes to manuscript:** P7, L1-11. Included description of visual stratigraphy in section 3.4.

P3 - L25 - Reword from "third spot" to "third profile".

> Changed accordingly.

P3 - L28/29 - Use of the term "clear" here needs further clarification. Again - highlights need to include a section (maybe initially in section 2) that provides a more detailed description of the basal ice facies observed and clarification of the significance of the use of the term "basal".

> We clarified that "clear" here refers to being entirely bubble-free. We have included a full description of the visual stratigraphy in section 3.4.

**Changes to manuscript:** P7, L1-11. Added full description of the visual stratigraphy to section 3.4.

P4 - Figure 1 - Include scale in Figure 1A. P4 - L4 - "20cm vertical intervals"

> The original sketch in Figure 1A was not to scale. However, we have added a new Figure 1 showing the glacier site (an a zoom-in on the tunnel location) as orthophotos, thus including GPS coordinates (Swiss grid) for scale.

**Changes to manuscript:** Added new Figure 1 with orthophotos.
P5 - Section 3.2 - Where the stable isotope measurements taken from all the
ice blocks? (Fig 2 suggests not)
Yes, in fact isotope data from block 2-5 in profile 2 is missing. The other
profiles have continuous isotope measurements (at least one measurement
per block). We have added information to the text to clarify this.
**Changes to manuscript:** P6, L2. Added Statement to clarify no data is available for
block 2-5.
P5 - L3 - "The outermost 10 cm of each block exposed to the tunnel was removed"
Changed accordingly.
**Changes to manuscript:** P5, L7-8.
P5 - Section 3.3 - Which blocks were used for the radiocarbon dating?
We provide this information in Table 1, first column. We have slightly
rearranged the text in the column to clarify this, now separating block number
and combustion temperature.
**Changes to manuscript:** Table 1
P6 - Section 3.4 - Include a description of the macroscopic characteristics of the ice
facies investigated here - ideally refer to an ice facies classification scheme.
Explain why the clear ice facies was specifically targeted for analysis.
We appreciate the suggestion and have added a new paragraph to this section
describing the macroscopic characteristics of the ice facies. We have also
added a new table (Table 2) to summarize the characteristics following the
classification scheme of Hubbard et al. (2009). The clear ice facies was not
specifically targeted, but rather a result of the search for different basal ice
characteristics, i.e. visual differences w.r.t profile 2.
**Changes to manuscript:**
•  P7, L1-11: Added new paragraph.
•  Added new Table 2.
P6 - Section 4.1- It's worth emphasising here that the measured temperatures are
significantly lower than those previously recorded by Haeberli.

  Thank you, we added a respective remark to this section. We suspect this may
  be connected to the artificial cooling installed in recent years.

**Changes to manuscript:** P8, L14: Added statement.

P7 - L7 - Equilibriation?

  Yes, referring to the time needed for the sensors to be in equilibrium with the
  ambient ice temperature.

**Changes to manuscript:** P8, L19: "limited time for establishing equilibrium".

P7 - Section 4.2 - Explain the significance of a replication of the basal isotope
anomaly. Does this indicate that the basal ice formed from precipitation during
colder climatic conditions?

  The significance of refinding the basal isotope anomaly lies in the fact that this
  supports the view of the basal ice not having undergone substantial changes
  over the last 25 years (i.e. since the anomaly was first described by Lorrain
  and Haeberli (1990)). We state this on page 9, Lines 11 ff. (revised
  manuscript). We also state that a full investigation of the origin of this anomaly
  is beyond the scope of this work. That said, as already discussed by Lorrain
  and Haeberli (1990), and also by Keck (2001) and Wagenbach et al. (2012), a
  pure atmospheric origin of this signal is very unlikely, with post-depositional
  processes probably contributing to this signature. We have added a more clear
  reference to this circumstance.

**Changes to manuscript:** P9, L8-9. Added statement.

P8- Figure 2 - Illustrate which samples have been obtained from the clear ice (cf.
Figure1D/E).

  Changed accordingly. Used grey shading in what is now Figure 3.

P8 - Again, a brief description of the ice facies and their key characteristics (e.g.
debris content and bubble content) would help provide a context for the
microstructural characteristics.

  As discussed above we have followed this valuable suggestion and included a
  facies description in section 3.4 and Table 2, to which we again refer to here,
  especially regarding the clear basal ice of profile 1.

P9 - Figure 3 - Where have have these results been obtained from? "Selected results"
rather than "exemplary results".

| | |
|---|---|
| 268 | Changed accordingly, we now provide this information in the caption of Figure |
| 269 | 4. |
| 270 | |
| 271 | P9 - Section 4.4 - Does progressive downwasting and thinning of the ice provide a |
| 272 | potential explanation for the fall in temperature? |
| 273 | |
| 274 | Interesting suggestion- after some consideration we would rather expect that |
| 275 | thinning of the ice would allow atmospheric temperature variability to |
| 276 | penetrate further into the ice, hence probably more likely associated with |
| 277 | warming than cooling. Assuming that Lorrain and Haeberli measured the |
| 278 | temperature only in the tunnel, not deeper in the walls of the ice (like we did), |
| 279 | the two sets of reported temperature are not straightforward to compare. |
| 280 | As we discuss in the manuscript, we believe that the englacial temperature |
| 281 | measured in our vertical boreholes are to some extent the result of the |
| 282 | artificial cooling of the tunnel. Another potential effect would be changes in |
| 283 | surface energy balance by fabric cover and snow plowing (ski resort). |
| 284 | However, it is difficult to disentangle these anthropogenic technical measures |
| 285 | from natural effects. Without detailed measurements of energy fluxes, |
| 286 | including latent fluxes, and refreezing/sublimation at the tunnel walls during |
| 287 | several years, the potential effects cannot be quantified and explanations |
| 288 | remain uncertain. |
| 289 | The certain and important consequence in the context of our work, however, |
| 290 | is that the ice remains frozen to bedrock thus far. |
| 291 | |
| 292 | P9 - L6 - Suggest rewording to - "...from basal temperature measurements revealing |
| 293 | temperatures well below the pressure melting point" |
| 294 | |
| 295 | Changed accordingly. |
| 296 | |
| 297 | P10 - L6 - "karst" rather than "carst" |
| 298 | |
| 299 | Changed accordingly. |
| 300 | |
| 301 | P10 - L19/20 - Emphasise why a cold-based thermal regime is required |
| 302 | to preserve old basal ice. |
| 303 | |
| 304 | Changed accordingly. P13, L11ff. |
| 305 | |
| 306 | P10 L19-22 - What is the source of this organic material and how has it been |
| 307 | incorporated into the basal ice? |
| 308 | |
| 309 | This is of course important- the organic material is assumed to be of eolian |
| 310 | origin originally deposited on the glacier surface (like the dust-type layers |
| 311 | visible in the stratigraphy). The basal layer, which contains a substantial |
| 312 | amount of sediment from the bed, has been avoided for 14C analysis. We have |
| 313 | included a statement to make this clear. |

**Changes to manuscript:** P13, 17-18
P10 - L26-28 - What is the layering composed of? Variations in bubble content and
ice crystal size? Suggestion here would be that this could be foliated ice in which
case it could be illustrative of shear deformation.
We have clarified this and again referred to the new table with the ice facies
characteristics. Given the regular nature of the layering parallel to the bed we
interpret the layering as not supporting any signs of folding or stratigraphic
disturbance, at least for the situation at profiles 1 and 2.
**Changes to manuscript:** Changes in section 3.4 and 4.3.
P10 - L29 - What is meant by the use of the term "dark". Bubble free?
The term "dark" refers to the visual appearance of the layer with respect to the
ambient ice. However, it is a result from larger amounts of dust in this layer,
i.e. not from being bubble-free. We have clarified this in the new Table 2 and in
section 3.4.
**Changes to manuscript:** P7, 6-7
P12- L6-7 - Earlier text suggested limited deformation whilst this suggests the
potential for deformation - watch for possible contradictions here. Would typically
expect significant shear strains in the basal ice layers of non-temperate glaciers
frozen to hard rock beds.
Thank you for pointing out this possible ambiguity in reading this section,
which we were not aware of. We state "that a moderate deformation is to be
expected" in section 3.4 and, in this paragraph, that our findings in ice
microstructure do not contradict the presence of shear deformation resulting
from cold-based conditions. Having said that, in discussing the observed
vertical gradient in age we are concerned with shear-introduced layer
thinning, not turbulent ice flow. The latter is clearly not supported by the
visual stratigraphy. We also discuss that the vertical age gradient could, as an
alternative, also result from glacier growth interrupted by phases of
stagnation or ablation and ultimately suggest further investigation. In this
view, we do not see contradictions- however, we have reworded this section
slightly to make our view more clear.
**Changes to manuscript:** P15, L1 ff.
P12 - L24-25 - Key finding - Give greater emphasis within the section.

Thank you for this suggestion. We have added text and elaborated on the
paleoglaciological and paleoclimatological perspective on our key finding.
As this work has been designed as a pilot study, final paleoclimatic
interpretations will greatly benefit from a larger sample of measurements at
various sites. The focus of this paper is to demonstrate the unchanged
existence of the lowermost layers and their potential for drawing conclusions
with the methods demonstrated applied to more locations.
**Changes to manuscript:** P15, L18-24 and P15, L28-32.
P13 - L5 - Suggest re-wording to: "Temperature measurements demonstrate basal
temperatures that are well below the pressure melting point..."
Changed accordingly.
P13 - L9 - "...five ice blocks suggests a chronological order..."
Changed accordingly.
P13 - L14-15 - This final sentence is the key finding. Give greater emphasis (new
paragraph?) and elaborate briefly on the potential palaeoclimatological and
palaeoglaciological implications.
We have restructured this into a separate paragraph to further emphasize the
significance of our key finding. In our view the main message is that although
one site can only provide limited direct paleoclimatic insight, we have
demonstrated the potential when extending this approach to other sites with
greater geographic coverage in the Alps.
**Changes to manuscript:** P16, L15 ff.
References Hubbard, B. & Sharp, M., 1995. Basal ice facies and their formation in the
Western Alps. Arctic & Alpine Research, 27(4), 301-310. Knight, P.G., 1997. The
basal ice layer of glaciers and ice sheets. Quaternary Science Reviews, 16(9), 975-
993.

---

## Author Comment (AC3) · 19 Dec 2017

**"Investigating cold based summit glaciers through direct access to basal ice: A case**

**study constraining the maximum age of Chli Titlis glacier, Switzerland" by Pascal**

**Bohleber et al.**

- Response to reviews -

*Please note:*

• *Author's responses to the referee's comments are in blue*

• *Changes in the corresponding revised manuscript are highlighted in red*

• *All line numbers in "Changes to manuscript" refer to the new revised version*

• *All new references can be found in the new manuscript*

**Response to referee #3, B. Hubbard**

General comments
The paper reports ice temperatures, stable-isotope concentrations and radiocarbon
dates from ice located near the base of Chli Titlis, Switzerland. The manuscript
reports that the basal ice is cold (and infers that this has been the case for a
considerable period of time) and that it has a maximum age of _5000 years BP. I find
the manuscript to be of interest, but also that it still needs fairly substantial revision
and improvement in a few areas. My main concerns relate to two aspects of the
manuscript.

We thank the referee for his valuable comments and suggestions, which we
believe have helped us significantly to improve the revised version of the
manuscript. The work has been designed as a pilot study focused on
investigating the unchanged existence of the lowermost layers and their
potential for drawing conclusions regarding the maximum age of ice at the site.
The main concept was to assess the feasibility of the approach before being
applied to more locations in the future. Since the authors lack a background in
geology we particularly appreciate the suggestion to include a description of
the ice facies.  We have added this aspect to the revised manuscript and will
also include the ice facies investigation in our set of methods for future work at
Chli Titlis and other sites.
A more detailed response to the referee's comments in presented below.

1. The measured ice temperatures at the location are too close to the glacier's
surface and/or the access tunnel to be able to ignore these external influences.
Seasonal variations in surface temperature typically penetrate 10 to 15 m into the
ice. Further, temperature within the tunnel is artificially influenced and, to some
extent, controlled (as clear from p. 7 paragraph 1). The reported temperatures are also not substantially below freezing, so error 'estimated at 0.2 degrees C' really needs a more formal quantitative analysis and accompanying statement. I recommend plotting the thermistor time series to reflect ice cooling back down (presumably exponentially) to host temperature following borehole-wall warming by the steam drilling. I believe this aspect of the manuscript needs greater focus and for all the relevant information to be brought into one early section. For example, published temperatures are reported towards the base of page 9 that should really form part of this background material. Temperature control in the tunnel is also mentioned elsewhere later in the manuscript.

> We fully agree with the referee that both the seasonal variations and the artificial cooling need to be considered in interpreting the measured temperatures. In fact we tried to emphasize this important point already in the original manuscript. Regarding the thermic disturbance by stream drilling and subsequent equilibration, we have ensured to wait (40-60 mins) long enough for temperature fluctuations to be well below the measurement accuracy. Regarding the latter we refer to the study by Hoelze et al. (2011) that employed identical sensors.
>
> From the referee's comment we understand that there is the need to present this information in a more concise way and early in the manuscript. We have modified the text and added to section 3.1 accordingly. We are continuing the discussion with the referee regarding thermal conditions by responding to other comments made by the referee below.

**Changes to manuscript:** New paragraph starting out section 3.1 to summarize the settings of the ice tunnel relevant to temperature.

2. The manuscript reports on the temperature of basal ice, but presents no formal analysis of the nature that ice. Such an analysis would be useful both for the reader to understand the nature of the environment, and because certain physical basal-ice facies are indicative of certain basal processes and conditions. I recommend the revised manuscript include a formal analysis of the types of ice present at the sample locations (see review by Hubbard and others (2009) in Quaternary Science Reviews and references therein).

> We thank the referee for bringing this to our attention. We consulted the suggested literature and now present a full description of the visual stratigraphy at the three sampling sites in the ice tunnel. We also include a summarizing table with an ice type classification, adopting the scheme of Hubbard et al. (2009). We believe that by this means, the similarity between profiles 1 and 2, but also their difference regarding the basal ice are clarified. This also concerns differences with respect to profile 3 in the far end of the tunnel.

**Changes to manuscript:** Added to section 3.4 "Visual stratigraphy and physical ice properties". Added Table 2 including an ice facies description.

Specific comments (page/line):

1/1 "with great success" adds little and is a value judgement; I recommend deletion.

Changed accordingly.

1/3 "...low altitudes may also contain old ice if locally frozen..."

Changed accordingly.

1/5 "However, with recent warming and consequent glacier mass loss,..."

Changed accordingly.

1/6 Delete ", however," and "Since sampling and dating the lowermost ice usually requires..."

Changed accordingly. We also generally tried to improve the second part of the abstract and reworded accordingly.

1/8 "We combine standard glaciological tools..." (and such 'tools' should just be specified as the term is rather too open)

Changed accordingly.

1/9 "...physical properties and radiocarbon dating."

We now use the term "state-of-the-art micro-radiocarbon analysis" in order to distinguish it from conventional radiocarbon dating (the application to glacier ice being a challenge not least due to the low carbon concentrations).

1/11 this "pioneering exploration" needs to be specified for the definite article ("the") replaced. (Note: I have not gone through the rest of the manuscript in the same detail; the grammar can still be improved)

We now use the indefinite article. Thank you for your help in improving the language. We have tried to also improve the grammar of the rest of the manuscript.

2/15 These statements make clear the need for a formal analysis of the ice types present and sampled for this study.

As stated above, we have followed the referee's suggestion and now include information on the ice types (cf. new Table 2).

2/23 What is "glaciological surveying"?
We clarified this by giving examples (mass balance measurements, ground-
penetrating radar) to the tools used in the cited study.
**Changes to manuscript:** P2, L26. Please note that we have rearranged this part of
the Introduction in an effort to improve clarity.
3/9 "... generally seem to be low" ideally needs some specification and
quantification.
We agree that it would be preferable to include a quantitative statement here.
However, we can only refer to what is reported in the cited study by Haeberli
et al. (2004), which does not provide more detail in this respect.
3/18 - 21 I recommend combining this material with temperature data from the
existing literature in order to present as complete and accurate a situation as
possible relating to the thermal history of this site. That this ice is, and has been,
cold is central to the manuscript's message.
Thank you. We follow this suggestion and have integrated the text in the
revised section 3.1.
**Changes to manuscript:** Moved to new paragraph in revised section 3.1.
5/8-14 I think the manuscript would benefit from a more formal statement of
isotopic error here. Currently, some delta D values are issued with caution because
of 'large uncertainty'. I would prefer to see formal error bars added to each data
point.
After considering this remark and also the comments made by the two other
referees we have decided to include a detailed plot showing the co-isotopic
data of profile 2, for which reliable delta D measurements were available. We
decided not to consider the delta D data of profile 1 further, in view of the
large uncertainties involved. However, we now clarify this and also report the
respective measurement uncertainties.
**Changes to manuscript:** P6, L5-7, added text. Added Figure 4.
7/25 Tes, there appears to be a pattern here that broadly matches one(s) recorded
elsewhere, but such a comparison should include all other profiles (including ice
coring literature from Alpine glaciers at least) so the reader is convinced that this
particular pattern is significantly over-represented. Also, if it is real, the explanation
is a little truncated. Could it be related to the formation of clear facies basal ice by
deformation induced preferential expulsion of light isotopes?

We appreciate this comment and take this as encouragement for further
investigation into the origin of the basal isotope anomaly (which is in fact
ongoing work).  We will especially also consider the hint to deformation-
induced preferential expulsion of light isotopes. The detailed explanation of
the isotope anomaly was not part of the study presented here, however, and is
certainly an intricate matter that deserves a separate investigation. An
overview of the present state-of-the-art regarding the isotope anomaly is
presented in Wagenbach et al. (2012) to which we have little to add at this
stage. We have tried to point out that, within the present work, we are merely
using the anomaly as a marker for the basal ice, previously described by the
earlier study of Lorrain and Haeberli (1990).
We have added an additional statement to clarify that the anomaly is not
regarded as being a climatic signal of atmospheric origin.
**Changes to manuscript:** P9, L8-9.
7/33-35 I'd like to see this co-isotopic plot (including error bars). Which of the less
certain delta D values were used and what is their associated error. If the data are
not of sufficient quality to 'interpret in more detail' then they may not be of
sufficient quality to present at all; at present, the reader cannot judge this.
Thank you for this suggestion. As mentioned above after careful consideration
we decided to i) clarify the associated errors, ii) not consider the delta D
values of profile 1 further due to high measurement uncertainty and iii) show
the co-isotopic data of profile 2 in a new Figure (Figure 4). The latter provides
additional overlap with the previously reported co-isotopic analysis by Lorrain
and Haeberli (1990), which we now include in the discussion.
**Changes to manuscript:**
•   P6, L5-7. Added text regarding measurement uncerainties.
•   Added Figure 4 with co-isotopic data.
8/5-8 Is it possible to illustrate these crystal size differences and the sub-grain
boundary and elongation conditions mentioned? The elongated crystals sound like
'interfacial' facies ice, agreeing with the congelation origin advanced in the
manuscript. The text states 'grain size' – which is presumably 'ice crystal size'.
We agree that the elongated crystals and almost bubble-free conditions at the
base of profile 1 point towards congelation ice (and state this accordingly). At
the same time we are not entirely sure how to best illustrate the crystal size
differences other than reporting them in the text.
**Changes to manuscript:** P10, L8. We have also clarified the meaning of "grain size"
being equivalent to "ice crystal size".

9/Fig. 3 These images are not very clear and seem to give no indication of scale
We had to reduce the size of the images in order to keep the file size
manageable. We have increased the image quality and also include an
indication of scale.
**Changes to manuscript:** Revised Figure 5, improved image quality and included
scale.
9/2-4 This reference to possible warm temperatures in the past seems at odds with
the general thermal interpretation of the site as cold. Perhaps some text could be
spent on rationalizing these seemingly contrasting thermal conditions.
We agree about this apparent contrast and in fact our main intention was to
discuss in this paragraph that the cold-based conditions are not immediately
intuitive. Connecting atmospheric temperature to the thermal conditions at
the base of the glacier is not straightforward of course and would require a
detailed investigation of the surface energy balance. The latter is highly
complex, not least due to the anthropogenic technical measures (ski area). We
already discuss the role of recent negative mass balance and surface covers
above the cave changing radiative fluxes, but also snow accumulation (wind
drift) and percolation of meltwater and rain. We have taken the referee's
comment as a suggestion to add some more details, including the apparent
challenges involved in surface energy balance conditions.  A thorough
calculation of past and present changes in energy fluxes governing the thermal
regime is not yet feasible within the scope of this manuscript, but will be
subject to future investigations starting with the installation of a monitoring
network – which hopefully will be funded.
**Changes to manuscript:** P12, L12-14. Explicit reference to surface energy balance
and technical measures adding to the complexity of the thermal conditions at Chli
Titlis
10/6 'karst'
Changed accordingly.
10/9-10 More here of relevance to the thermal conditions
Please see our comments made above and below.
10/17 True, but to focus on the future does not address the issue raised by this
paragraph – that there may be issues complicating the temperatures reported. . .
Surface temperature changes should be accounted for in any interpretation of point
temperatures recorded within a thickness of _ 10m. Same for the tunnel, although this zone of influence is likely smaller because the temperature changes in the
tunnel will presumably be muted. Are there records of external temperature at the
surface (or nearby, to which a lapse rate can be added) and in the tunnel?
To continue the above discussion, we agree that with an ice thickness of less
than 10 m, it cannot be ruled out that the base is affected by seasonal
temperature variations. We now mention this circumstance more explicitly.
There is a weather station operated by MeteoSwiss at the telecomunication
tower on Titlis glacier, close to the ice cave. Over the last decade, the data
shows a typical seasonality ranging between 5 and -15°C (monthly data) and
an annual mean temperature around -3 to -4°C  (e.g. -3.5, -2.9 and -3.7 for
2014,15,16, respectively). Unfortunately, to our knowledge no temperature
logging is available for the tunnel.
That said, the englacial temperature profile is determined by additional factors
(which we are sure the referee knows), especially regarding conditions at the
surface (e.g. meltwater, snow cover,...) which today are highly disturbed by
anthropogenic technical measures (surface covers, reworking of the snow
surface, etc.). Multi-annual logging of englacial temperature by installing a
thermistor chain in a borehole would be needed in this context. Although this
clearly went beyond this initial study at Chli Titlis, we have already
incorporated this aspect in our follow-up studies at other sites in the Eastern
Alps- featuring both borehole temperature logging and automated weather
station data. We also take this as encouragement for future work at Chli Titlis.
In conclusion to this discussion, and to reiterate our previous statement, we
fully agree that under these circumstances finding sub-zero ice temperatures
is not trivial and has to be considered in view of the current technical
measures by the ski resort.  We appreciate the discussion and have tried to
elaborate on these important issues even more in the respective paragraph.
**Changes to manuscript:**
• P11, L12-13. Explicit reference to influence of seasonal temperature
variability.
• P12, L18. Mention reworking of snow surface for ski area maintenance.
• P13, L8-9. Suggest future logging of englacial temperature at the site.
11/Fig. 4 Some of the structures here do seem to indicate ice deformation. How does
this relate to the interpretation of generally undeformed ice in this location? Maybe
these features are not deformation structures, but some analysis and interpretation
might help address this possible issue.
We are not entirely sure which of the structures the referee refers to but are
trying to give an answer as we understand the situation. Except for the basal
layer, we interpret the visual layering in the ice as originating from the surface,
e.g. dust or soil material being deposited and accumulated on the glacier and
subsequently incorporated into the ice body. We do not think that deformation is entirely absent (cf. also section 4.3). However, we observe no evidence of
turbulent ice flow or macroscopic layer folding. In addition, some localized
basal melting may have occurred, and could have contributed to the observed
lateral differences in age structure between profiles 1 and 2. We have added
these considerations to the revised text.
**Changes to manuscript:** P15, L1ff. Added to discussion of deformation and vertical
age gradient.